# Successfully Recruiting Adults with a Low Socioeconomic Position into Community-Based Lifestyle Programs: A Qualitative Study on Expert Opinions

**DOI:** 10.3390/ijerph17082764

**Published:** 2020-04-16

**Authors:** Josine M. Stuber, Cédric N. H. Middel, Joreintje D. Mackenbach, Joline W. J. Beulens, Jeroen Lakerveld

**Affiliations:** 1Department of Epidemiology and Biostatistics, Amsterdam Public Health Research Institute, Amsterdam UMC, VU University, De Boelelaan 1089a, 1081 HV Amsterdam, The Netherlands; 2Upstream Team, www.upstreamteam.nl, Amsterdam UMC, VU University, De Boelelaan 1089a, 1081 HV Amsterdam, The Netherlands; 3Athena Institute, Faculty of Science, VU University Amsterdam, De Boelelaan 1085, 1081 HV Amsterdam, The Netherlands; 4Julius Center for Health Sciences and Primary Care, University Medical Center Utrecht, Utrecht University, Universiteitsweg 100, 3584 CG Utrecht, The Netherlands

**Keywords:** recruitment, lifestyle intervention, socioeconomic status, qualitative research

## Abstract

We explored experts’ perceived challenges and success factors in the recruitment of adults with a low socioeconomic position (SEP) for participation in community-based lifestyle modification programs. We conducted semi-structured interviews with 11 experienced project coordinators, based on a topic list that included experiences with recruitment, perceived barriers and success factors, and general views on recruitment strategies. Results revealed challenges related to the context of the program (e.g., limited program resources), psychosocial barriers of the participants (e.g., mistrust or skepticism), practical barriers (e.g., low literacy or having other priorities), and reasons to decline participation (e.g., lack of interest or motivation). Success factors were related to securing beneficial contextual and program-related factors (e.g., multi-layered recruitment strategy), establishing contact with the target group (e.g., via existing networks, community key-members), methods to increase engagement (e.g., personal approach and involvement of the target group in the program process) and making participation easier (e.g., providing transport), and providing various types of incentives. Concluding, the group of participants with low SEP covers a wide spectrum of individuals. Therefore, multiple recruitment strategies at multiple layers should be employed, and tailored. The lessons learned of those with hands-on experiences will help to enhance recruitment in future programs.

## 1. Introduction

Community-based lifestyle programs aim to create healthy community environments through systemic changes in health promotion approaches [1]. The effectiveness of such programs and the potential to evaluate them in scientific studies highly depends on reaching the target population and meeting the estimated sample size to provide adequate power for evaluation of research findings.

Adults with a low socioeconomic position (SEP), mostly characterized by a below average occupational position, educational level and/or income [2], are often considered a “hard-to-reach” population. Response rates of programs–whether or not in the context of a scientific study–are typically lower in groups with a low SEP than those with a high SEP.

Some non-responder surveys of observational programs report that populations with a lower SEP more often refuse to participate; up to 15% more as compared to populations with a higher SEP [3,4,5,6,7]. Recruitment for lifestyle programs focusing on the adoption and maintenance of healthy lifestyle behaviors is even more challenging than recruitment for observational programs. Groups with a low SEP are often underrepresented and generally show higher attrition rates—roughly 30% as seen in some studies [8,9,10]. As such, health promotion approaches that are based on the results of these programs are likely to fit less well with the needs of the underrepresented groups. At the same time, the evidence base consistently shows that groups with a low SEP are at higher risk of cardiometabolic diseases and cardiovascular events [11,12], so their underrepresentation in health promotion approaches hinders opportunities to decrease socioeconomic inequalities in health [13].

Reasons for lower participation and retention rates of populations with a low SEP may be attributable to the typical highly agentic nature of lifestyle behavior programs. They do not align with the fewer resources, poor (health) literacy, low self-efficacy and emotional resilience and a less supportive physical and social environment that are more often present in this target group [14]. The resulting underrepresentation of low SEP respondents may cause selection bias, a decrease in the external validity of program findings [15], and waste of research resources as insufficient recruitment is often the main cause of early study termination [16].

Although insufficient recruitment of participants with a low SEP is a major issue, not much research has been conducted on effective recruitment strategies for this population. The available literature on successful recruitment strategies is often focused on clinical settings or evaluates recruitment in health system contexts, whereas evidence on effective strategies focusing on community-based lifestyle programs is limited, in particular among populations with a low SEP [17,18,19].

Nevertheless, some systematic reviews have summarized the recruitment challenges and/or success factors among hard-to-reach groups in health-related research programs in general [19,20,21,22]. Main reasons for non-participation that were identified were that the target group had a lack of awareness about health-related research programs, did not understand the program information, the process or significance, and that they did not understand the need for randomization. Other reasons were mistrust and perceptions that participation may cause harm, stigma, or exploitation. Moreover, some individuals believed that participation presented no personal benefit to them. Other barriers were the lack of a supportive community, deteriorating health, lack of transportation options, low literacy, and financial or social problems. In turn, strategies to enhance program enrolment rates were: Engagement of local peers, family or known community members, the use of familiar locations, social relationship building, and providing easy to read program information. Social marketing, such as mass mailing, posters and media outreach, was described as effective strategy to initiate contact with the target group—often combined with community outreach methods. Other components for successful recruitment were providing incentives and developing community connections.

To date, the research on effective recruitment strategies mostly focused on the (potential) participant perspective rather than on the researchers’ or program leaders’ perspective. Data on the perspective of experts who have been involved in ongoing or completed community-based lifestyle programs targeting populations with a low SEP may provide important additional insights. These experts might be able to share experiences with various recruitment strategies which can support new program leaders in efficiently developing new recruitment strategies for this target group. With a qualitative approach, we therefore aimed to identify experts’ perceived challenges and success factors contributing to successful recruitment of participants with a low SEP into community-based lifestyle programs.

## 2. Methods

### 2.1. Setting and Recruitment of Experts

We searched *PubMed* and the database of *Loket Gezond Leven* of the Dutch National Institute for Public Health and the Environment for community-based lifestyle programs in The Netherlands in order to identify relevant experts. Further identification was done through the networks of researchers in the field of healthy lifestyle research. Community-based lifestyle programs were defined as programs conducted in a community setting. We excluded programs which involved recruitment via general practitioners or community-based clinics as those are important gatekeepers for hard-to-reach populations. Recruitment without the involvement of these health care workers has shown to be more challenging [19].

We focused primarily on lifestyle programs targeting sufficient physical activity and/or a healthy diet, considering those lifestyle behaviors as major contributors to chronic diseases and socioeconomic differences in health [11]. Experts had to have first-hand experience with the recruitment of participants with a low SEP into community-based lifestyle programs, completion of the recruitment period should not be longer than approximately five years ago, and experts had to be either an academic researcher or an academically trained employee of a municipal organization or other relevant expertise center. Identified experts were invited for study participation via e-mail.

### 2.2. Interviews

The interviews were either held face-to-face in a private room at the respondents’ office, or by telephone. Interviews were guided by a semi-structured interview protocol (Table 1). The first part of the interview addressed their experiences (challenges and success factors) with recruitment in a recently completed or current program, with an emphasis on the challenges and the success factors. The second part addressed their general views on recruitment strategies, based on their own experiences. The interviews lasted between 30 and 45 min and were audio-recorded. Recordings were transcribed verbatim excluding unrelated social talk. A member check was performed by inviting respondents to provide commentary on the transcript. All data were collected between April 2019 and July 2019.

The study was conducted in accordance with the Declaration of Helsinki and was approved by the ethics committee of the Amsterdam UMC—location VUmc. All respondents were informed on the study procedures by a standardized information letter. Written informed consent was obtained prior to the interviews. The reporting here follows the *Consolidated criteria for reporting qualitative studies* (COREQ) 32-item reporting guideline [23].

### 2.3. Data Analysis

All identifying information was removed from the transcripts prior to data processing. Transcripts were imported in Atlas.ti (ATLAS.ti Scientific Development GmbH, Berlin, Germany) [24] for analysis. Data were first analyzed with a deductive content analysis approach, followed by an inductive grounded theory approach [25]. Transcripts were closely read to become familiar with the content, and primary coded and analyzed by the first author, according to an iterative process and semi-open coding technique. As a means of triangulation, the coding of each quote was critically evaluated and discussed with the second author until consensus was reached.

The initial codebook included pre-identified themes that reflected potential challenges in recruiting hard-to-reach groups extracted from literature [20,21]. To structure the challenges and label them with predefined sub-themes, a supporting theoretical framework was used. As the decision for program participation could be viewed as a participant behavior—influenced by various psychosocial factors—we used the Theory of Planned Behavior (see Appendix A) as theoretical basis of the initial codebook [26]. Predefined sub-themes were labelled and categorized using the theory’s three domains as overarching predefined themes: *Attitude*, *Subjective norms*, and *Perceived behavioral control* (Appendix A). The three domains were presumed to affect the intention to enroll and/or the actually decision to enroll in a research program: (i) *Attitude*: referring to the attitude, perceptions and knowledge of health research–this component included sub-themes on the lack of understanding and awareness on health research, mistrust, and social incompatibility with the researcher; (ii) *Subjective norm*: including lack of social support, seeing no personal benefit in participation and lack of motivation and interest, and (iii) *Perceived behavioral control*: covering lack of opportunities to participate, low literacy, and having other (conflicting) priorities in life.

The initial codebook primarily focused on the recruitment challenges. Strategies addressing these challenges (i.e., the success factors) identified from the interviews were categorized using the similar three predefined themes from the initial codebook. Themes and sub-themes within the challenges as well as within the success factors were then further specified through open coding and a constant comparative analysis. After fully coding the first five interviews, themes and sub-themes were critically examined and revised where necessary. The last six interviews were coded with the use of this updated codebook. Changes in the codebook were logged during the coding process and over 90% of the final sub-themes were identified after coding the first five interviews. Barring revisions of codes, no new sub-themes emerged from the last six interviews and data saturation seemed to have been accomplished.

### 2.4. Research Team

The first author (female, predoctoral fellow in the research field of lifestyle epidemiology) conducted all interviews. The first author had a prior work-related relationship with one of the interviewees. All others were informed on her scientific background and her personal motivations for conducting this work at first contact—namely the preparation for a new lifestyle program [27]. The second author is a predoctoral fellow (male) conducting research in the field of systems innovation and transition theory using qualitative research methods. Among the other members of the research team are two postdoctoral fellows with previous experience in conducting qualitative studies and one professor (one male, two females) all focused on the research field of lifestyle epidemiology.

## 3. Results

We invited 12 experts of whom 11 responded and volunteered to participate. The interviewed experts worked at a university or an academic hospital, expertise center on health disparities, or the community health services (Table 2). The majority of the programs that were conducted by the experts focused on improving healthy dietary behaviors and/or physical activity levels, and investigated their effectiveness in experimental study designs. Two of the included experts held an academic master degree, two were predoctoral fellows, six were postdoctoral fellows and one was a professor. Experts did not provide feedback on the current study findings, only on the transcript of their interview.

A structured overview of key challenges and success factors in the recruitment of participants with a low SEP is presented in Table 3. The themes and sub-themes within the challenges and the success factors were deductively obtained from the interviews. Next, we inductively identified a hierarchical structure between the themes, with themes on the higher levels forming prerequisites for subsequent themes.

Regarding the recruitment challenges, overarching challenges at the first level were the contextual challenges related to the program. The second level of challenges consisted of psychosocial barriers among potential participants towards health-related programs and/or program leaders. Practical barriers towards participation formed the third level of challenges. The fourth level of challenges consisted of reasons to decline program participation.

Among the success factors, the first level for successful recruitment included securing beneficial contextual and program-related factors. Second, approaches to establish contact with the target group became relevant. Once contact is established, methods to increasing engagement for participation were identified as third necessary condition for successful recruitment. Having achieved sufficient engagement, factors on the fourth level focused on making participation as easy as possible. The final level in a successful recruitment strategy related to providing incentives.

### 3.1. Challenges

Four major levels in recruitment challenges were identified, as further detailed below.

#### 3.1.1. Contextual Challenges

This first level of challenges, the contextual challenges, relate to the program and the program leaders themselves. Experts experienced limitations in time or available resources, which some attributed to the fact that programs were part of a doctoral thesis. As such, program conception up to evaluation had to fit within its allocated four-year period which is common for a Dutch doctoral trajectory. The program design itself may also present challenges, as the use of scientific procedures such as the informed consent procedure or (large) questionnaires is often unavoidable. As one interviewee described:

“Simply, as a researcher you also have to account for these things [laborious scientific study procedures, e.g., informed consent]. (…) But it also relates to the fact that lengthy questionnaires are often needed in studies—you don’t make friends with that either.”(R1)

The lack of capacity and expertise for planning an effective recruitment strategy presented another challenge, possibly causing unrealistic expectations towards the recruitment process:

“In hindsight, [thinking we could recruit within] half a year was too optimistic. A number of issues played a role there. First, our team lacked experience with recruiting this target group at the time of writing the recruitment plan. Second, it [the program] was part of a PhD program which had to fit within the allocated four years.”(R1)

Study fatigue among the target group was mentioned as challenge, as some groups or specific neighborhoods were frequently approached for a variety of programs.

#### 3.1.2. Psychosocial Barriers Towards Health-Related Programs or Program Leaders

The second level of identified challenges related to psychosocial barriers among the potential participants towards health-related programs and program leaders. The experts noted that there was a lack of experience with scientific study procedures among this target group, as participants sometimes sign up for the program but reconsider their decision once they understand what kind of procedures are involved in scientific practices. The complexity of questionnaires and the level of detail used in questionnaire items were noted as a threat to program inclusion and retention rates as well. One interviewee explained:

“Sometimes questions were just too difficult or at a level of detail that people cannot connect with at all. They really do not understand that it is all necessary.”(R4)

Experts noticed there can be mistrust or skepticism towards the program leaders and the program procedures. They explained this by the number of organizations a potential participant often already has to deal with, such as municipalities, debt restructuring, youth care or food banks. Consequently, program participation is simply too much for them, or prior negative experiences with these organizations affect their decision to enroll in a new program. Hesitation among potential participants could also complicate recruitment according to some interviewees, as participants may be unsure about the program expectations towards them or are afraid to be judged on their answers to questions. Furthermore, experts experienced that their program information not always reached the target group as planned. Despite efforts by the experts to aim for non-stigmatizing content, media editors sometimes use inappropriate reporting, and therefore, possible adversely affect potential participants’ perceptions of the study. Moreover, most strategies used to reach the target group do not allow monitoring of who received that information, and if the method of delivery was the most suitable. One of the experts warned for using intermediate persons in the recruitment as you lose control over the information dissemination and exact content:

“Placing recruitment responsibility with other people can be dangerous because you lose control over the information. They can say the craziest things.”(R3)

#### 3.1.3. Practical Barriers Towards Participation

The third level of challenges related to practical barriers to participate. Experts commonly attributed low program participation by this target group to their other (conflicting) priorities in life, and one expert said that the resistance towards program participation from family members resulted in non-participation. Some practical challenges were mentioned: Measurement moments take a lot of time and do not always match the expectations of the participant, or participants are frequently not showing up for appointments likely due to voluntary participation without obligations or consequences, or the lack of e-mail use among a part of the target group. As one interviewee said:

“For the lowest socioeconomic groups, sending e-mails does not work. Nine out of ten times they have no e-mail address or they do not read their e-mail.”(R9)

It was also noted there is a large group of individuals who are willing to participate in a program and initially seem eligible for participation, but ultimately are unable to understand methods or questionnaires due to a lack of basic education, illiteracy or a language barrier. Furthermore, limited options for transportation was another barrier towards participation: Car ownership is not common among this group and public transport use is frequently perceived as too expensive. Lastly, some experts noted that health related issues such as an injury or social or psychological problems can make participation impossible.

#### 3.1.4. Reasons to Decline Participation

Reasons to decline program participation was the last level of identified challenges. Experts indicated that individuals could simply feel unmotivated for participation. The lack of motivation is especially challenging to address within a program, as the incremental effort put into convincing non-motivated participants often does not pay off (i.e., drop-out among non-motivated participants is higher). As one interviewee noted:

“My experience is that the harder you work to include individuals in your trial, the more people you lose eventually. So, then you actually invest a lot of time for nothing. That is a bit of a trade-of.”(R3)

Other reasons for lack of interest to participate were related to anxiety, for example with regard to going someplace new to do something new (e.g., sports), or shame about their personal life. Individuals can also perceive participation lacks personal benefit or feel their participation holds little value. They may perceive themselves as not important, or what do they do as not important enough. Some individuals can even feel exploited:

“They view themselves as cash cow.”(R4)

### 3.2. Success Factors

Five major level of success factors in the recruitment of participants with a low SEP were identified, as further detailed below.

#### 3.2.1. Program Context and Actors

The first level of success factors was related to the program context and the actors related to it. When designing a new program, there are multiple factors that can be addressed to acquire beneficial contextual factors in order to reach the target group.

Experts highlighted that the program design itself also influences the recruitment possibilities. They explained that the first group of included participants generally consists of intrinsically motivated and innovative participants (“early adaptors”) who are not likely be representative for the entire population. Selective recruitment was viewed as highly problematic when recruiting for observational studies, where the representativeness of the study sample is a crucial element. However, selective recruitment of motivated participants in randomized controlled trials (where the main goal is to test a certain hypothesis in a controlled setting) was viewed as worthwhile in order to reach the estimated sample size. This semi-selective inclusion of motivated participants has the additional benefit that it can reduce attrition rates during follow-up.

Experts also proposed the adaptation and tailoring of standard questionnaires or instrument to the target group as program-related success factor. The major advantage is reducing participant burden, though adapting instruments should not result in a major loss of validity. Investigating the validity of the adapted instrument should therefore be considered, according to the interviewees.

The perception and attitude of the program leader towards the challenges that are encountered during recruitment was viewed as important. Experts indicated that for successful recruitment, a program leader needs a realistic and open view towards the recruitment process and should apply a flexible pragmatic approach, for example by postponing program recruitment deadlines. Also, according to experts it is crucial to learn from the target group, by trying to view from their perspective and identifying what is logical and familiar for them. This information can be used to tailor the recruitment strategy.

Some experts pointed out to strategically have chosen a recruitment location where they knew their target population was present. Moreover, the use of multiple recruitment strategies was viewed as essential, as different individuals will react to different kind of strategies. At the same time, this will create opportunities for repeated exposure to the program which in turn will enhance familiarity with the program as well.

Investigating the current or previous values within a community, or the contextual issues faced, was another success factor. Taking into account the context in which the target group lives will enable opportunities to connect with the target group. As one interviewee said:

“I think it is important that you also know the history of the neighborhood and what is going on at that moment for those living in the neighborhood.”(R6)

Favorable circumstances were mentioned which included building upon previous programs as this may warrant a solid foundation, and using nationwide news articles as this may have a major influence on the popularity of a program. Nevertheless, enabling such favorable circumstances cannot always be planned:

“That [news item on a large national news website] was of course highly successful. That did really well. Though you cannot enforce something like that.”(R2)

#### 3.2.2. Methods to Reach the Target Group

Using various methods to reach the target group was identified as a second level of success factors. All experts viewed the use of existing networks as a highly important step. Networks primarily included organizations that already successfully reached the target group, including municipal organizations, social services, health brokers, community centers, food banks, clothing banks, patient or elderly unions, social housing corporations, neighborhood representative organizations, voluntary organization, charities, and schools. Existing networks could also be hobby related, such as sports clubs or gardening groups.

Almost all experts viewed the visitation of the location of the target group as an effective strategy, as well as the involvement of known and trusted community key-members in order to mobilize the target group. These key-members are likely to be embedded within existing participant networks, such as social workers, general practitioners, community ambassadors, or local residents. Especially for the lowest socioeconomic groups, help of these key-members was considered indispensable:

“If you really want to reach the most vulnerable groups, you cannot just enter the arena like that. An intermediary person is crucial.”(R4)

The use of (social) media and contacting local newspapers was also viewed as potentially supportive for recruiting participants, although with similar limitations. Some experts proposed that, rather than recruiting high numbers of the lowest socioeconomic participants, this method could help to create awareness on the program as part of a multi-layered recruitment strategy. Some experts used the word-of-mouth method. Neighborhood mail-out was effectively used by two experts. However, it was noted this strategy is not likely to result in the recruitment of the lowest socioeconomic groups.

#### 3.2.3. Ways to Increase Engagement

The third level of identified success factors related to ways to increase engagement. According to all interviewees, the most important strategy to achieve this was to make yourself as a program leader known to the target group and have a personal approach. This familiarizes potential participants with the program employees and provides them with the opportunity to ask questions. This establishes the feeling of safety and increases engagement–both components that enhance program participation:

“I think personal and informal contact is the most important factor for successful recruitment. Once they have met you, as a researcher, barriers towards participation are much lower.”(R7)

Frequently mentioned was identifying participants’ potential personal benefits with regard to participation. As one interviewee pointed out:

“Ask yourself: What could be your target groups’ ‘wow-factor’ with regard to participation?”(R2)

Once identified, experts indicated this information can also be used to manage expectations towards the program to prevent disappointment, as some participants have different expectations of the program than what will actually be provided. Furthermore, almost all experts stressed that in order to reach engagement the target group should be involved in the program process. Identification of the target group needs with regard to development of a new program is an example of this. Moreover, experts indicated a top-down approach should be avoided:

“The point is that you have to connect. You have to listen to your target group and you have to synchronize with them. It doesn’t matter what someone’s background is like, as long as you communicate with and involve the target group.”(R6)

Relationship building goes beyond having a personal approach. Some experts stressed that long-term relationship building is required when aiming to include to lowest socioeconomic groups. This includes building trust through reoccurring encounters between program employees and potential participants, before inviting individuals to participate in the program.

Clear and honest communication on program aims and outcomes and the value of participation was a frequently mentioned strategy to increase engagement. Another mentioned strategy to enhance trust was encouraging participants to be program ambassadors and to recruit others from their personal network. Last, avoiding questions on sensitive topics was also reported to create trust. An example was to avoid asking participants about their income levels when targeting a population with low SEP, with as alternative strategy recruiting in low SEP neighborhoods or other locations such as at the food bank.

#### 3.2.4. Making Participation Easier

The fourth level of identified success factors relate to making program participation easier. Reducing the required effort and costs which are related to participation as much as possible was by the experts seen as the most important factor. Strategies they used were providing (or avoiding the need for) transportation, matching those participants who are willing to share transport, sending reminders for appointments, and adjusting time schedules to participants needs.

Providing understandable program material and tailoring of communication styles were other strategies to make program participation easier. Experts explained they used visual communication such as images or films to grab attention and avoid the need for written communication. One interviewee referred to a video they made that illustrated someone from the target group receiving a recruitment letter in the mail while remarking the importance to participate. This video was showed after a church service.

Several experts provided support and guidance during questionnaire completion or used easy to understand language in the questionnaire as much as possible. Experts approaches for tailoring material to the target group and gender, such as limiting (home)work and individual assignments, taking into account cultural differences and language barriers, using repetition of information in communication (materials), and pre-evaluating program materials by the target group.

A number of experts referred to shortening the time-span between recruitment and actual program participation including data collection as an important strategy to reduce required effort and increase retention rates between inclusion and the first measurement:

“We decided that interested participants could start immediately [with the program], because once you have them interested—you don’t want to make them wait.”(R1)

Another strategy was to combine the program with existing groups of the target population who already have a fixed (e.g., weekly) meeting moment, with examples such as language courses, coffee or tea meetings in a community center, or a walking group. The pre-existing social relationships between participants, the familiarity with the location and familiarity with the time point as a result of this existing groups were viewed as major advantages:

“Most groups have a fixed moment in the week when they have a coffee together. You could request to extend that moment [to use for you program].”(R5)

#### 3.2.5. Incentives for Participation

The fifth level of identified success factors related to providing incentives for participation. Experts mentioned three types of incentives, namely a social incentive (e.g., group activity or doing something for their community), a personal insensitive (e.g., health check), or a financial incentive (e.g., coupons). According to some of the interviewees, combining several types of incentives is the most optimal to facilitate enrolment within a program:

“The group activity was something we considered as an advantage of our intervention. The social aspect thereof. We noticed participants were going to pick each other up for the meeting to go together.”(R1)

## 4. Discussion

Through the qualitative exploration of expert opinions, we identified a hierarchy of challenges and success factors related to the recruitment of participants with a low SEP into community-based lifestyle programs. Findings confirm, and extend previously described challenges and success factors when recruiting hard-to-reach groups [19,20,21,22]. The expert views on recruitment methods specifically targeting individuals with a low SEP into community-based programs revealed that the use of existing networks and a personal approach were seen as key-elements for successful recruitment.

### 4.1. Multi-Layered Recruitment Approach

Along with the use of existing networks and a personal approach, experts viewed the involvement of community key-members, visitations of the location of the target group, involvement of the target group into the program processes, and reducing the efforts and costs related to participation as most important strategies for successful recruitment of groups with a low SEP. However, the generalizability of all identified successful strategies is challenging and no single approach can be applied to all programs. This is partly due to the heterogeneity between programs and communities, and partly due to the heterogeneity within this target group itself [28].

It appeared that it is unlikely that all individuals will respond equally to the applied recruitment strategies. Via passive strategies (i.e., methods where potential participants are expected to sign themselves up for participation, such as mass mailing) it is likely more challenging to reach those with the lowest educational level, low literacy or language barriers. They will probably be more prone to respond to active approaches (i.e., where the program employee initiates contact with the potential participant). Yet, passive strategies may serve to recruit sufficient participants. The scientific literature describes effective use of passive strategies for recruitment of groups with a low SEP into health-related research programs, where such strategies often yield more than two third of the study sample [19,22].

The effectiveness of passive approaches such as mass-mailing is likely attributable to their wide reach. Although their absolute yield is high, the inclusion rates among all who are reached is frequently low. Active approaches are likely more suitable to result in a high inclusion rate among all reached. Furthermore, solely using passive strategies might lead to larger risk of general selection bias. Underrepresentation of those with the lowest educational level, low literacy or language barriers is likely to be introduced. As such, the adoption of a comprehensive and multi-layered recruitment approach combing passive with active approaches is crucial to reach all individuals within the group with a low SEP—to be able to evaluate program effectiveness and ultimately decrease socioeconomic inequalities in health [13,29,30,31].

### 4.2. Underlying Barriers Hindering Successful Recruitment

The experts perceived underlying barriers related to the scientific research system or the capacity of the program leaders, which appear to hinder successful recruitment of groups with a low SEP.

Some experts acknowledged a misalignment on the educational level between researchers and their target group. Such “ivory-tower biases” could threaten successful recruitment, as it makes tailoring of program materials more challenging. It is therefore highly recommendable to consult the target group for feedback and recommendations on the planned program materials and strategies.

Adherence to the Good Clinical Practice principles is universally recognized as requirement when conducting health-related research programs [32]. Local ethics committees monitor programs based on these principles. One of their monitoring tasks is to ensure potential participants are well-informed prior to signing participation consent. In practice, experts experienced that these regulations often limit their abilities to create context-specific recruitment materials tailored to a population with a low SEP, as they are obliged to use long and complex standardized information letters formats. Acknowledgement of the diversity of research programs and their target groups and guidance by the ethics committees for study-specific adaptation of materials would enable possibilities for improved tailoring of program materials. For groups with a low SEP, a suitable option could be guidance on visually presented information instead of the standardized letters [33].

Balancing feasibility and costs can be challenging considering programs often depends on external (public) funding to secure these posts. Consequently, there is often insufficient time, budget and staff allocated to the (extension of the) recruitment period or strategies. Moreover, in the Netherlands, it is common to investigate the effectiveness of lifestyle related programs as part of a doctoral thesis. Time is therefore a set factor—mostly at a four-year period—and predicting participant inclusion rates is extremely difficult. Considering all the research phases has to fit within these four years (i.e., planning a program, recruiting participants, conducting the program, collecting data and analyzing results) acceptance of disappointing recruitment rates is often perceived as unavoidable. When aiming for successful recruitment, carefully planning and evaluating recruitment strategies warrants higher levels of attention and devotion in the development phase of the program than it currently often receives.

### 4.3. Strengths and Limitations

This study has several strengths. We provide a novel contribution to the existing knowledge on factors related to recruitment of groups with a low SEP in community-based settings, by presenting a comprehensive overview of the challenges and success factors as viewed by experts. The lack of contradictions in the results between experts underlines the validity of our findings. Validity of the data was secured via the member check, and validity of the data analysis by discussing all codes with two authors with different scientific backgrounds (i.e., triangulation).

Some limitations should be noted as well. First, some of the interviews were conducted via phone which may have affected possibilities to build rapport with the expert as face-to-face interaction was not possible. All interviews were conducted by—and with—academics, and analyses and interpretation of the data were done by academics as well. The SEP of the researchers and interviewees was thus not aligned with those of the target group under study which again could have resulted in “ivory-tower biases”. Last, all experts’ experiences were related to the Dutch setting which limits global generalizability of the study findings.

## 5. Conclusions

The findings of this paper highlight the importance of a flexible program mentality and designing a multi-layered recruitment strategy. The use of existing networks and visitation of the target groups location are most frequently used methods to get in contact with the target group. Moreover, using a personal approach towards the target group, identifying potential personal benefit related to participation, and reducing effort and cost related to participation were by experts seen as key-elements for successful recruitment.

The group with a low SEP covers a wide spectrum of individuals among which not every strategy will be equally effective for all. The here described lessons learned of those with hands-on experiences can guide future program leaders and researchers when planning recruitment strategies for this specific target group and setting.

## Figures and Tables

**Table 1 ijerph-17-02764-t001:** Summary semi-structured interview protocol.

Introduction	
1	Scope and structure of the interview
**First part: Specific Program**	
2	General program recruitment information (e.g., target sample size versus included sample size, planned versus actual duration of recruitment period)
3	Applied recruitment strategies (e.g., amount and type of recruitment strategies, adjustment recruitment strategies during recruitment period, response to (individual) recruitment strategies, (un)expected (un)successful strategies, most (un)effective strategies, feedback by (potential) participants)
4	(Un)expected challenges (e.g., (un)expected challenges during recruitment, strategies to handle the (un)expected challenges and retrospective approach)
**Second Part: Program Leaders General View**	
5	General opinion on challenges and success factors, including recommendations for future researchers
**Final Part:**	
6	Additional points and closing interview

**Table 2 ijerph-17-02764-t002:** Experts’ work setting and focus of a recent or current program (*n* = 11).

Work Setting:	Focus of the Program:		
	Interventional Program	Observational Program	Implemented Program
**University or Academic Hospital**	Improving dietary behaviors and physical activity levels (*n* = 3)	Dietary behaviors (*n* = 1)	
	Improving physical activity levels only (*n* = 3)	Health literacy (*n* = 1)	
**Expertise Center on Health Disparities**		Smoking behaviors (*n* = 1)	
**Community Health Services**		Health monitoring (*n* = 1)	Improving dietary behaviors (*n* = 1)

**Table 3 ijerph-17-02764-t003:** Structured overview of key challenges and success factors identified from the interviews.

	Theme	Sub-Theme	Number of Times Mentioned (by Number of Respondents)
**Challenges**	**1. Contextual challenges**		49 (10)
		Limited program resources and time	12 (7)
		Barriers program leaders’ perception	8 (6)
		Program design	21 (5)
		Limited capacity program leader	4 (4)
		Overburdened target group	6 (3)
	**2. Psychosocial barriers towards health-related programs or program leaders**		36 (10)
		Lack of awareness or understanding health related program	16 (8)
		Mistrust or skepticism	11 (7)
		Incomplete/indiscreet program information dissemination	14 (6)
	**3. Practical barriers towards participation**		43 (11)
		Other priorities	13 (7)
		Practical challenges	12 (7)
		Language barrier	12 (6)
		Lack of transport or limited financial resources	7 (5)
		Low literacy or unable to understand methods/questions	7 (5)
		Health related issues	3 (3)
		Resistance by others	1 (1)
	**4. Reasons to decline participation**		16 (6)
		Lack of motivation and interest	7 (4)
		Shame or anxiety	6 (4)
		Seeing no (personal) benefit or value	4 (2)
**Success Factors**	**1. Program context and actors**		64 (10)
		Program design	23 (8)
		Program leaders’ perception and attitude	19 (6)
		Strategic location or setting	10 (5)
		Multiple strategies	9 (3)
		Context sensitivity	6 (3)
		Fortunate circumstances	3 (2)
	**2. Methods to reach the target group**		96 (11)
		Use existing networks	52 (11)
		Visit location target group (not via or with key-member)	21 (10)
		Involvement community key-members	32 (9)
		(Social) media channels	16 (4)
		Word of mouth	3 (3)
		Mail-out	2 (2)
	**3. Ways to increase engagement**		107 (11)
		Make yourself known and personal approach	32 (11)
		Identify potential personal benefit	30 (8)
		Ask and involve target group in program process	26 (8)
		Relationship building	21 (8)
		Communicate program outcomes and value	19 (8)
		Involvement target group in recruiting others	7 (5)
		Limiting sensitive topics	4 (3)
	**4. Making participation easier**		54 (11)
		Reduce effort and costs	18 (8)
		Tailor communication and material to target group and gender	12 (7)
		Make material understandable	14 (6)
		Short time-span recruitment and data collection	7 (4)
		Use existing groups of target population	8 (3)
	**5. Incentives for participation**		19 (9)
		Social incentive	9 (5)
		Personal incentive	6 (5)
		Financial incentive	9 (4)

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
