# Peer review of "Successfully Recruiting Adults with a Low Socioeconomic Position into Community-Based Lifestyle Programs: A Qualitative Study on Expert Opinions"

_ijerph, 2020, doi:10.3390/ijerph17082764_

Round 1

Reviewer 1 Report

In this paper, by conducting semi-structured interviews with a number of experienced project coordinators, the authors explored challenges and success factors in the recruitment of adults with a low SEP for participation in community-based lifestyle modification programmes. Results show multiple challenges related to the context of the programme, psychosocial barriers of the participants, practical barriers, and reasons to decline participation. In addition, success factors found in the study are associated with securing beneficial contextual and programme related factors, establishing contact with the target group, methods to increase engagement and making participation easier, and providing various types of incentives. Based on such results, it is recommended that multiple recruitment strategies at multiple layers should be employed and tailored. Overall, the problem being addressed is of significant interest to academic and social communities. Moreover, the article is well written and clearly laid out, and tables are informative. However, the quality of this paper can be further improved if the following point is addressed:

The authors should rank the key challenges and success factors identified from the interviews, so that future researchers would know which ones they need to prioritize.

Author Response

Reviewer #1

In this paper, by conducting semi-structured interviews with a number of experienced project coordinators, the authors explored challenges and success factors in the recruitment of adults with a low SEP for participation in community-based lifestyle modification programmes. Results show multiple challenges related to the context of the programme, psychosocial barriers of the participants, practical barriers, and reasons to decline participation. In addition, success factors found in the study are associated with securing beneficial contextual and programme related factors, establishing contact with the target group, methods to increase engagement and making participation easier, and providing various types of incentives. Based on such results, it is recommended that multiple recruitment strategies at multiple layers should be employed and tailored. Overall, the problem being addressed is of significant interest to academic and social communities. Moreover, the article is well written and clearly laid out, and tables are informative.

>          We thank the reviewer for these compliments.

However, the quality of this paper can be further improved if the following point is addressed: The authors should rank the key challenges and success factors identified from the interviews, so that future researchers would know which ones they need to prioritize.

>          We have reordered the sub-themes presented in Table 1 such that these are now prioritized based on number of respondents who mentioned the sub-theme. Likewise, we have adjusted the corresponding text flow in the results section. All changes made are highlighted in yellow (page 6-12). We did not adjust the theme order, since those were structured in a hierarchical order with themes on the higher levels forming prerequisites for subsequent themes (as described in lines 187-189, page 5).

Reviewer 2 Report

This paper contains important insights into getting low SEPs into community-based lifestyle programs. This is a useful study that will contribute to future practice.

Please refer to the following for minor comments.

1 Maybe subtitles are better with qualitative research on Expert opinions than Expert opinions.

2 In the introduction, the background of the research is described sufficiently logically.

3 The method is fully described with the necessary information.

4 The results section is fully documented with the required information.

5 In the discussion, the results are properly considered and the limitations of the study are clearly mentioned.

6 In the conclusions section, could you like to elaborate a little more on the results obtained in this study.

Author Response

Reviewer #2

This paper contains important insights into getting low SEPs into community-based lifestyle programs. This is a useful study that will contribute to future practice. Please refer to the following for minor comments.

1 Maybe subtitles are better with qualitative research on Expert opinions than Expert opinions.

>          We have clarified the title of the article as follows:

Successfully recruiting adults with a low socioeconomic position into community-based lifestyle programmes: A qualitative study on expert opinions’

2 In the introduction, the background of the research is described sufficiently logically.

3 The method is fully described with the necessary information

4 The results section is fully documented with the required information.

5 In the discussion, the results are properly considered and the limitations of the study are clearly mentioned.

>          We thank the reviewer for this positive feedback.

6 In the conclusions section, could you like to elaborate a little more on the results obtained in this study.

>          We have included additional key-results in the conclusion section (lines 518-527):

‘The findings of this paper highlight the importance of a flexible programme mentality and designing a multi-layered recruitment strategy. The use of existing networks and visitation of the target group’s location are most frequently used methods to get in contact with the target group. Moreover, using a personal approach towards the target group, identifying potential personal benefit related to participation, and reducing effort and cost related to participation were by experts seen as key-elements for successful recruitment.

The group with a low SEP covers a wide spectrum of individuals among which not every strategy will be equally effective for all. The here described lessons learned of those with hands-on experiences can guide future programme leaders and researchers when planning recruitment strategies for this specific target group and setting. ‘
